

# A modified seasonal cycle during MIS31 superinterglacial favors stronger ENSO variability

Flavio Justino[1], Fred Kucharski[2], Douglas Lindemann[1], Aaron Wilson[3], and Frode Stordal[4]

[1]Department of Agricultural Engineering, Universidade Federal de Vicosa, PH Rolfs, Vicosa, Brazil
[2]The Abdus Salam International Centre for Theoretical Physics, Trieste, Italy
[3]Polar Meteorology Group, Byrd Polar and Climate Research Center, The Ohio State University, Columbus, OH, USA
[4]University of Oslo,Department of Geosciences, Forskningsparken Gaustadalleen, Oslo, Norway

**Correspondence:** Flavio Justino (fjustino@ufv.br)

**Abstract.** It has long recognized that the amplitude of the seasonal cycle can substantially modify climate features in distinct timescales. This study evaluates the impact of enhanced seasonality characteristic of the Marine Isotope Stage 31 (MIS31) on the El Niño-Southern Oscillation (ENSO). Based upon coupled climate simulations driven by present day (CTR) and MIS31 boundary conditions, we demonstrated that the CTR simulation shows signicant concentration of power in the 3-7 year band and on the multidecadal time scale between 15-30 years. However, the MIS31 simulation shows drastically modified temporal variability of the ENSO, with stronger power spectrum at interannual time scales but absence of the decadal periodicity. Increased meridional gradient of SST and wind stress in the Northern Hemisphere subtropics, in concert with weaker seasonal cycle of the windstress in the MIS31 simulation, revealed to be the primary candidates responsible for changes in the equatorial variability. The oceanic response to the MIS31 ENSO extends to the extratropics, and fits nicely with SST anomalies delivered by paleoreconstructions. The implementation of the MIS31 conditions results in distinct global monsoon system and its link to the ENSO in respect to current conditions. In particular, the Indian monsoon intensified but no correlation with ENSO is found in the MIS31 climate, diverging from conditions delivered by our current climate in which this monsoon is significally correlated with the NINO34 index. This indicates that monsoonal precipitation for this interglacial is more closely connected to hemispherical features than to the tropical-extratropical climate interaction.





# 1  Introduction

The Marine Isotope Stage 31 (MIS31; early Pleistocene 1085-1055 ka) is a prime paleoclimate period to simulate and analyze the global environmental response to a significantly modified climate forcing (Lisiecki and Raymo, 2005; Yin and Berger, 2012a). This interval was characterized by boreal summer temperatures that were several degrees greater than modern climate,
with a substantial recession of the Northern Hemisphere (NH) sea ice (Melles etal., 2012; Justino etal., 2017).

On long time scales, Earth's climate is primarily controlled by external and internal processes related to the astronomical forcing and the atmospheric concentration of greenhouse gases (Stocker etal., 2013; Erb etal., 2015). Internal modes of climate variability, such as the El Niño-Southern Oscillation (ENSO), the Pacific Decadal Oscillation (PDO) and the Northern Annular Mode (NAM), also induce climate anomalies on interannual and decadal time scales (Bjerknes, 1964; Mantua etal., 1997;
Thompson and Wallace, 2001). Indeed, changes in the physical and dynamical characteristics of the ENSO have been related to seasonal and interannual global-climate distubances (Cai etal., 2014).

The impact of equatorial dynamics and ENSO have been found in different equilibrium climates forced by glacial and inter-glacial conditions (Karamperidou etal., 2015). For instance, palaeoreconstructions have demonstrated a significant reduction in the climate variability associated with ENSO during the mid-Holocene ($\approx$ 6000 years before present (BP); Karamperidou etal.
(2015)). The glacial maximum climate was affected by distinct ENSO variability as well; however, during this time the ENSO demonstrated larger-amplitude self-sustained interannual variations compared to current conditions (Tudhope etal., 2001; An etal., 2004; Toniazzo, 2006; Zhu etal., 2017).

Larger differences in the east-west SST gradient in the equatorial Pacific that began 1.17 million years ago, has also been claimed to support the onset and intensification of the modern Walker circulation (McClymont and Rosell-Melé, 2005). How-
ever, it can be argued that there is no a preferential dominant region in the equatorial Pacific, because global climate distur-bances have been found in response to NIÑO3, NIÑO4 or NIÑO34 anomalies. For instance, Yin etal. (2014) indicates that warmer conditions during the MIS13, an interglacial that occured at approximately 0.5 million years ago, in the Indian-Pacific warming pool, amplifies the insolation effect and contributes to a large increase of summer precipitation in southern China, whereas dryer conditions occur in northern China.

The far-reaching effect of equatorial dynamics on climate has been demonstrated by Karami etal. (2015). They argued that colder summer sea surface temperatures (SSTs) in the central tropical Pacific during MIS13 contributes to the strengthening of the northern Pacific subtropical high, increasing the transport of moisture into the East Asia Summer Monsoon (EASM). Moreover, they highlight the significant influence of the east-west SST differences in the tropical Pacific in maintaining the link between the tropical Pacific and EASM.

(Sun etal., 2010b) based on seven million years of wind and precipitation variability on the Chinese Loess Plateau, demon-strated that monsoonal fluctuations at orbital-to-millennial scales is dynamically linked to changes in solar insolation and in-ternal boundary conditions, which are tightly related to interglacial ENSO variability. Additional analyses by Rachmayani etal. (2016) demonstrated that the global monsoon system during interglacial stages differs from current conditions, which may be characterized by the out-of-phase between the West African and Indian monsoon.



Significantly modified periodicity and amplitude of past ENSO regimes and their global influence, shed light on the potential effect of human induced climate change on the equatorial Pacific, and consequently on future ENSO-like climate. Furthermore, it should be argued that disagreement in the magnitude of cooling or warming among coupled climate models and paleorecon-structions may be related to the local responses of temperature and precipitation elicited by distinct ENSO (Peltier and Solheim, 2004; Jost etal., 2005; Yin and Berger, 2012b; Dolan etal., 2015; Justino etal., 2017).

The effect of ocean dynamics also modify the tropical-polar interaction due to different ENSO flavors (Wilson etal., 2014, 2016), that through changes in the atmospheric circulation can result in anomalous sea-ice/ice sheet mass characteristics (Steig etal., 2013). The warmer climate of the MIS31 has been shown to result in an overall reduction of snow cover that to some extent, may be similar to estimates based on simulations of human-induced future global warming (Frei and Gong, 2005).

The understanding of the air-sea interaction related to the equatorial Pacific and its climate response at interannual and multi-decadal timescales in distict epochs, such as interglacial intervals, is therefore crucial. It is also vital to understand past interglacial intervals that are characterized by depleted ice sheets to verify the potential effects of increasing atmospheric $CO_2$, as the stability of the West Antarctic Ice Sheet (WAIS) will be a key climate factor in decades to come (Nicolas etal., 2017).

## 2 Coupled Climate Simulations

Two model simulations have been performed with the International Centre for Theoretical Physics - Coupled Global Climate Model (ICTP-CGCM; Kucharski etal., 2016). ICTP-CGCM consists of the atmospheric global climate model "SPEEDY" version 41 (Kucharski etal., 2006) coupled to the Nucleus for European Modelling of the Ocean v3.3 (NEMO) model (Madec, 2008) with the OASIS3 coupler (Valcke, 2013).

The atmospheric component runs at T30 horizontal resolution, and there are eight levels in the vertical. NEMO is a primitive equation z-level ocean model based on the hydrostatic and Boussinesq approximations. This version applies a horizontal resolution of 2° and a tropical refinement to 0.5°. The ocean component has 31 vertical levels with layer thicknesses ranging from 10 m at the surface to 500 m at the ocean bottom (16 levels in the upper 200 m). Additional details on the ICTP-CGCM coupled model are discussed by Justino etal. (2017). Farneti etal. (2014) used the ICTP-CGCM to examine the interaction between the tropical and subtropical northern Pacific at decadal time scales, suggesting that extratropical atmospheric responses to tropical forcing have feedbacks onto the ocean dynamics leading to a time-delayed response of the tropical oceans.

Two simulations are evaluated: the modern climate driven by present-day boundary conditions (CTR) and a second experiment for the MIS31 forcing (see Fig. 1 of supplementary material by Justino etal. (2017). The CTR (MIS31) simulation was run to equilibrium for 2000 (1000) years and the analyses discussed herein are based on the last 500 years of each simulation. The MIS31 run starts from equilibrated CTR conditions, including modifications of the WAIS topography based on Pollard and DeConto (2009), and the planetary astronomical configuration of 1.072 Ma according to Coletti etal. (2014).

The implementation of MIS31 Antarctic topography differs from the CTR counterpart primary by the absence of the WAIS, which according to Pollard and DeConto (2009), was induced by changes in ocean melt via the effect on ice-shelf buttressing



that coincides with strong boreal summer insolation anomalies. In all experiments, the $CO_2$ concentration was set to 325 ppm which is based on boron isotopes in planktonic foraminifera shells for the MIS31 interval (Honisch etal., 2009). The MIS31

and CTR experiments have been described in further detail elsewhere by Justino etal. (2017), but a brief discussion of the global climate differences between these two runs are provided below.

Table 1 shows the global and hemispheric surface temperature values for the CTR and MIS31 simulations, HadCRUT4 (Morice etal., 2012) for the base period 1961-1990 and ERA-Interim (ERA-I; Dee etal., 2011) for the 1980-2010 interval. Our CTR climate is warmer than the HadCRUT4 and ERA-I but better correspondence is found with the ERA-I. Larger differences

are noticed for the NH summer when the CTR climate is 2°C (1°C) warmer than HadCRUT4 (ERA-I). These differences arise from higher temperatures over land, because the SST and sea-ice distributions in the CTR simulation fit well with the ERA-I dataset, as shown by Justino etal. (2017). Temperature differences between the MIS31 and the CTR show that most of warming occurs in the boreal summer, reaching 1.2°C in the global mean, 2.2°C in the NH, and 0.4°C in the Southern Hemisphere (SH). Lower temperatures are demonstrated during DJF in the MIS31 run compared to the CTR simulation,

clearly showing the hemispheric seesaw effect of the astronomical forcing.

Due to astronomically driven reduced seaice, larger changes are located in the NH extratropics (see Justino etal. (2017)). It has to be mentioned that differences between the MIS31 and CTR simulation deliver negative surface temperature anomalies over southern Asia, western equatorial Pacific and South Atlantic. Compared to tropical and extratropical paleoreconstructions, the MIS31 simulation performs well with values that departure from paleoproxies by ± 1°C in the tropics between 20°N-3°S

(McClymont and Rosell-Melé, 2005; Medina-Elizalde etal., 2008; Herbert etal., 2010b, c; Li etal., 2011; Russon etal., 2011; Dyez and Ravelo, 2014) by up to -3°C within 41-67°N belt (Raymo etal., 1996; Herbert etal., 2010a; Li etal., 2011; Naafs etal., 2013) and ± 1.5°C in the SH between 23-42°S (McClymont etal., 2005; Crundwell etal., 2008; Scherer etal., 2008; Naish etal., 2009; Martínez-Garcia etal., 2010; Russon etal., 2011; Voelker etal., 2015).

These changes in the the atmosphere thermodynamics also review that the MIS31 climate is dictated by an enhanced seasonal

cycle compared to present climate (not shown). The inclusion of distinct astronomical forcing leads to NH peak summer (July) insolation, with an opposite effect in the SH, due to the interterhemispheric seesaw relationship of the precession cycle (Scherer etal., 2008; Erb etal., 2015). Zonally averaged, the MIS31 climate is remarkably warmer than the CTR during JJA except poleward of 45°S. During DJF, the MIS31 is slightly cooler between 45°S and 50°N (not shown).

The inclusion of MIS31 boundary conditions also results in changes in SLP and the vertical structure of the atmosphere.

Figures 1a,b show the eddy SLP (SLP with the zonal mean removed, $SLP_e$) and the Z200 (geopotential height at 200 hPa with the zonal mean removed). This strategy is important, because changes in circulation are dictated by changes to the gradient of geopotential rather than absolute magnitude anomalies (He etal., 2018). At the surface, $SLP_e$ anomalies exhibit an increase in the western North Pacific subtropical high in opposite to the drop in the eastern North Pacific (Fig 1a). This partially supports previous results by Mantsis etal. (2013), who found a large strengthening and a northward and westward expansion of the

northern Pacific summer anticyclone, driven by changes in the timing of perihelion.

According to Cook and Held (1988) and Timmermann etal. (2004) the meridional circulation $v\prime$ is proportional to the mean westerly circulation $u > 0$, which is also modulated by the seasonal cycle of the SLP. In the upper troposphere (200 hPa), this





induces southward anomalies over the eastern Pacific and northward and low-pressure anomalies on the downstream side of the Tibetan plateau in MIS31 (Fig. 1b); hence, weakening the jet stream in the MIS31 climate compared to the CTR counterpart.

This vertical structure with baroclinic anomalous pattern in particular over East Asia and western Pacific may be related to the ENSO dynamics in the MIS31 climate, as will be verified later.

These changes in the stationary wave induce substantial modifications in the windstress and SST/near surface air temperatures features delivered by the MIS31 climate. Indeed, Fig. 1c depicts warmer SSTs in the northeast and subtropical Pacific but cooler temperatures in the west Pacific. These changes along the equatorial belt are primary induced by weaker northeast trade

winds that reduced evaporative cooling and lead to less vigorous equatorial upwelling between 0-20°N. Moreover, windstress changes in the eastern Pacific reduce the cold tongue strength (Figs. 1c,d).

The ICTP-CGCM properly reproduces the equatorial thermocline depth (using the depth of maximum vertical temperature gradient) compared to the Levitus dataset (Levitus etal., 2000). The MIS31 forcing leads to a shallower thermocline and reduction of its zonal gradient (Fig. 1d), which is primarily related to the anomalous wind flow (e.g., Zebiak and Cane,

1987; An etal., 1999). A deeper thermocline however, is observed in part of the NIÑO3 region (Fig. 1d, contour). In the eastern Pacific, thermocline dynamics have been associated with changes in SST, the air-sea coupling, and ENSO (Leduc etal., 2009; Yang and Wang, 2009). This implies a weaker Walker circulation during the MIS31 interval that is supported by SST reconstructions (from Ocean Drilling Program sites 849, 847, 846, and 871) in the western and eastern equatorial Pacific (McClymont and Rosell-Melé, 2005).

Over the western Pacific, stronger equatorward winds (Figs. 1c,e) lead to cooler SSTs and enhanced subtropical cell, in concert with an intensified subtropical gyre (Figs. 1g,h). The wind-driven circulation may be evaluated by the Sverdrup transport defined as:

$$\psi(x) = \frac{1}{\beta\rho} \int\limits_{x_e}^{x} \frac{\partial\tau_x}{\partial y} dx \qquad (1)$$

where $\beta$ is the meridional derivative of the Coriolis parameter, $\rho$ is the mean density of sea water, and $\tau_x$ is the zonal

component of the wind stress. The integral is computed from the eastern to the western boundary in the North Pacific using modeled atmospheric wind stress data. The ICTP-CGCM model simulates the Sverdrup transport quite well (Fig. 1g) compared to the magnitude of the Sverdrup transport estimated from the International Comprehensive Ocean-Atmosphere Data Set (ICOADS; Woodruff etal., 2011).

The intensification of the Sverdrup transport by up to 6 Sv between 20-40°N in the Kuroshio region induces negative SST

anomalies due to the intrusion of colder sub-surface water related to the speed up of the subtropical cell (Fig. 1h). These processes are in phase with increased precipitation in the central Pacific region, but dryer conditions are noted in the Warm Pool region (Fig 1f). The convergence of wind anomalies (Fig. 1c) also indicates westerly flow with potential implications for the ENSO dynamics (Eisenman etal., 2005). In the Atlantic Ocean, warmer conditions are noticed in most of the NH which are tightly related to orbitally driven wind anomalies. While the northern Pacific shows a baroclinic structure, the Atlantic shows a barotropic pattern demonstrated by the $SLP_e$ and Z200 (Figs. 1a,b).





## 3  Harmonic analyses of MIS31 and CTR climates

Additional evaluation on modifications of the annual and semi-annual oscillation are provided below through harmonic analyses. The first order harmonics of meteorological parameters show long-term effects, while higher order harmonics represent the effects of short-term fluctuations that characterize different climate regimes and transition regions. Using this mathematical approach allows the identification of dominant climate features in the space–time domain desintangling small and large-scale processes driven by distinct periodicity (Justino etal., 2010, 2016).

Changes in the harmonic variance and amplitude are highly correlated with the amount of incoming shortwave radiation (SSR) in the MIS31 climate, as shown by differences in the $1^{st}$ harmonic (Fig. 2a-f) of SSR, SST, SLP and the HF. It has long been recognized that the equatorial climate exhibits an annual component which strongly dominates SST, windstress, and precipitation (Li and Philander, 1996, 1997). Nevertheless, the western equatorial Pacific and to a lesser extent the western Atlantic temporal variability present largest variance in the semi-annual component ($2^{nd}$ harmonic). The semi-annual component is strongly influenced by ocean-atmosphere interactions, in which the surface atmospheric flow and SST, feedback on the cloud structure further modifying the SSR and oceanic heat flux.

Figure 2a-d reveal that the semi-annual component, which is dominant in the western equatorial Pacific under current conditions (not shown), is weaker in the MIS31 climate allowing larger variance in the annual harmonic. This is highlighted in particular by SST and SLP distributions which potentially impact on ENSO characteristics (2b-d). It is also shown that similar patterns are displyed by the SSR and HF, and SST and SLP harmonics. The former (SSR and HF) experiences an interhemispheric distribution whereas the latter i(SST and SLP) is dominated by an equatorial east-west dipole in the Pacific.

Figures 2e-h show changes in the amplitude of the $1^{st}$ harmonic between the MIS31 and CTR simulations. The main features are shown as larger (smaller) NH (SH) amplitudes deliverd by the MIS31 run, in particular along the continental margins and in the Warm Pool/western Pacific area. Insofar as the western Pacific changes are concerned, it has been found that the local increase in windstress during (JJA) driven by the seasonality of SLP over central and western Pacific, are in concert with the higher SSR, SST and HF amplitudes. These changes in seasonality dramatically alter the MIS31 climate compared to the CTR climate in both spatial patterns and the main mode of variability (further discussed below).

This structure is not seen in the equatorial Atlantic where variance differences between the MIS31 and the CTR are meridional. In fact, under CTR conditions this can be interpreted as the tropical Atlantic variability (TAV) related to the continental monsoon forcing, windstress and air-sea interaction (Deser etal., 2010). However, due to orbitally-driven changes in SSR (Fig. 2a), the MIS31 climate in the tropical Atlantic shows weakening of the annual component southward to 10°N, and an intesification of the semi-annual oscillation between 10-20°N compared to the CTR run (Fig. 2b).

The SLP differences are more complex, showing a pattern that differs from zonal or meridional features (Fig. 2c), even though they are correlated with SSTs in the western Pacific (Warm Pool region). In the Atlantic, the $1^{st}$ harmonic weakens, allowing for sub-seasonal temporal variability (lower order harmonics) enhanced nearby the African coast (Fig. 2c).



## 4    MIS31 - Temporal and spatial characteristics of ENSO

It is expected that those changes in the atmospheric zonal and meridional circulations and the wind-driven oceanic flow can result in modifying ENSO frequency and power. The following explores the influence of the MIS31 forcing on ENSO indices. Among several mechanisms related to ENSO dynamics, the magnitude of the seasonal cycle in the equatorial region characterizes its onset, intensity, and frequency (Liu, 2002a; Nonaka etal., 2002; Timmermann etal., 2007). It has been argued that in case of strong seasonal cycle, the ENSO signal can be locked in phase and frequency with this external forcing, thus reducing its magnitude. The ENSO signal may also differ in strength and influence if computed over distinct oceanic regions, such as those defined by NIÑO3, NIÑO34 or NIÑO4 (Wilson etal., 2014, 2016).

Figures 3a,b show the ENSO power spectrum computed for the NIÑO34, NIÑO3, and the NIÑO4 using the Hadley Centre Sea Surface Temperature data set (HadISST; Rayner etal., 2003) and the CTR simulation. This is achieved by applying the Multi-Taper method (MTM; Thomson, 1982) a technique that has been demonstrated to fill limitations of conventional Fourier analysis.

All periodicities mentioned below are significant at the 95% confidence level. Compared with the power spectrum delivered by the HadISST, the ICTP-CGCM shows sharper peak in the 3-7 year band for all NIÑO indices (Fig 3a). The HadISST spectrum shows maximum concentration of power spanning the 3-5 year period. Under CTR conditions, significant concentration of power is also dominant on the multidecadal time scale between 15-30 years. Similar periodicity has been previously found by Nonaka etal. (2002). (Nonaka etal., 2002) attributes the equatorial decadal variability to the influence of winds in the trade wind bands which modifies the strength of the sub-tropical cell. It is interesting to note that NIÑO34, NIÑO3, and NIÑO4 differ in reproducing the decadal frequency, weakest in the NIÑO4. The HadISST does not show any periodicity on decadal time scales; even though the SST data spans 1870 to 2016, the length of the timeseries does not seem to capture this lower frequency. It has to be mentioned that Rodgers etal. (2004) claims that the zonal asymmetry related to decadal variability in the HadISST observations is weaker and not as regular as for instance in the ECHO-G model.

The weakening of decadal variability in the NIÑO4 region may be related to wind variability in the off-equatorial tropics as proposed by Nonaka etal. (2002). This assumption has been verified by computing the correlation pattern associated with the NIÑO indices. It turns out that the NIÑO4 relationship with the zonal windstress within 10-30°N is considerably weaker than that of NIÑO34 or NIÑO3 (not shown). Moreover, this weaker correlation between the NIÑO4 and windstress is not confined to the equatorial region but extends to the extratropics.

The incorporation of MIS31 boundary conditions drastically modifies the temporal variability of the interglacial ENSO (Fig. 3c). This simulation shows stronger power spectrum at interannual time scales between 3-7 years. Evaluation of the main causes related to the strengthening of the interannual variability in the MIS31 climate compared with the CTR counterpart is not straitforward. It has been found that an increased meridional gradient of SST and wind stress in the NH tropics (Fig. 2a), as simulated by the MIS31 run, may lead to stronger interannual equatorial variability in the MIS31 climate (Liu, 2002a, b; Erb etal., 2015). Likewise, the weaker seasonal cycle of the windstress in the MIS31 simulation may lead to stronger ENSO power at 3-7 years (Chang etal., 1994). The CTR climate SOI power spectrum also shows enhanced power at similar



frequencies found for the NIÑO34 and the NIÑO3 indices. This is in line with the spectrum of equatorial winds (0-20°N) that

shows enhanced power also at interdecadal time scales (Fig 3d).

The opposite is delivered by the MIS31 simulation, a fact that usefully serves to support the assumption of weaker decadal air-sea interaction during this interglacial. Indeed, for the MIS31 simulation, correlation values between the NIÑO34 index and the $1^{st}$ principal component (PC1) of windstress computed at 0-20°N are very low; whereas for the CTR run, these values are 0.6 when all timescales are included and 0.4 for conditions in which frequencies below 10 years have been filtered

out. Though previous studies have claimed that the equatorial Pacific interannual variability is primary forced by equatorial windstress (Nonaka etal., 2002; Timmermann and Jin, 2002), and the decadal variability is strongly connected to the off-equatorial windstress, our results show that the atmospheric flow between 0-20°N can induce decadal variability.

In fact, the decadal variability found in the CTR NIÑO34 power spectrum fits nicely with the proposed mechanism raised by Farneti etal. (2014). The SST anomalies at the equator induce changes in the windstress curl over the western Pacific, that

generate SST anomalies fluctuating on decadal time scales through tropical-subtropical interactions. Individual analyses to verify the roles of the North and South Pacific in inducing the decadal variability, demonstrate that most changes of power can be explained by the NH contribution. Interestingly, the NIÑO34 and windstress anomalies between 20°S-0 are highly anti-correlated with values of about 0.6 in both simulations. However, only in the MIS31 climate the windstress spectrum does exhibit enhanced interannual variability, indicating that the MIS31 ENSO dynamics are also driven at the 3-7 year period by

the SH flow. A fact that is not seen for the CTR climate.

Turning to the regression patterns induced by the NIÑO34 indices, Figure 4 shows that our coupled model reproduces the main tropical SST response to NIÑO34 (Fig. 4a), compared for instance with Cai etal. (2015). The patterns are displayed as amplitudes by regressing hemispheric anomalies on the standardized first principal component time series. The intensification of the NIÑO34 signal does not project substantial change in SST, though in the western Pacific, anomalies between ± 0.3°C

are noted (Fig. 4b).

The impact of NIÑO34 on SLP (Fig. 4c) extends globally and is fairly reproduced by the ICTP-CGCM compared to the National Centers for Environmental Precition - National Center for Atmospheric Research (NCEP-NCAR) reanalysis (Ji etal., 2015). The zonal dipole results from the contribution of the baroclinic component over the eastern Pacific and barotropic component over western Pacific, both related to the SST anomalies (Figs. 4c, a). The MIS31 NIÑO34 weakens the barotropic

and baroclinic patterns of the SLP as shown by the differences in the MIS31-CTR regression (Figs. 4c,d). In the equatorial region, the anomalies are related to the intensified winds nearby the Warm Pool region but weakening mid-latitude westerlies in both the northern Pacific and Atlantic (Figs. 4e,f). In the following, we compare temperature differences between the MIS31 and CTR to compiled data by Wet etal. (2016), but with focus on the ENSO responses (Table 2).

This is achieved by comparing the modeled SST anomalies for JJA to SSTs differences between the MIS31 and CTR delivered by the regression pattern related to the NIÑO34 index (ΔT). Differences between the reconstructions and the NOAA Extended Reconstructed SST V3b/ERA-I are also shown ($\Delta T_{re}$). Overall, model results and reconstructions agree indicating that the ENSO works in line with the astronomical forcing, inducing warming ($\sqrt{}$ in Table 2) however in some cases it acts in

5 the opposite ($X$ in Table 2).



## 4.1 Global and monsoonal precipitation

As shown in Figure 1f), it is evident that most wet and dry conditions in the MIS31 compared to the CTR are in agreement with the anomalous temperature pattern and diabatic heating, in line with ENSO-related precipitation (Dai and Arkin, 2017). Exception is found over southern Asia that experiences more precipitation despite the drop in temperatures, which may indicate the contribution of extratropical large-scale atmospheric dynamics.

To further investigate the MIS31 climate features it is evaluated the correlation between precipitation computed over regional Asia monsoonal domains, as defined by Yim etal. (2014) and the NINO34 index. The domains are: the Asia monsoon (AM, 10°N-45°N, 70°E-150°E); Australia monsoon (AUS, 5°S-20°S, 110°E-150°E); East Asia monsoon (EA, 22.5°S-45°N, 110°E-135°E); western north Pacific monsoon (WNP, 12.5°N-22.5°N, 110°E-135°E) and Indian monsoon (IN, 10°N-30°N, 70°E-105°E).

Estimates of changes in precipitation for past interglacials are still scarce, but our MIS31 simulation agrees with other studies showing enhanced Asia summer monsoon (Fig. 5a), during interglacials (An, 2000; Sun etal., 2010a). Enhanced seasonality and greater annual values of precipitation have also been documented across the East Russian Arctic in line with our modeling experiment (Melles etal., 2012). This is also true for the Asian monsoon insofar as seasonality is concerned (Fig. 5a). Controversy is raised by Oliveira etal. (2017) who argue that reduced seasonality in precipitation along western Mediterranean region during MIS31 leads to forest decline. Reduced seasonality during the MIS31 in that region is not supported by our MIS31 climate simulation, which in fact shows an increase in both summer and fall/winter precipitation (not shown).

Turning in particular for the monsoonal domains, it is clear the weakening of the relationship between the NINO34 and the Asia and Australia monsoonal domains during the MIS31 as compared to the CTR counterparts (Fig. 5). It should be mentioned that the equatorial Pacific seems to have a direct influence on the AM, WNP and the AUS monsoon precipitation (Fig. 5a,b,e). This is not the case for the East Asia and Indian monsoon. Under current conditions, the NINO34 is negatively correlated with monsoonal indices with values by up to -0.5 for the AM and AUS, in which the NINO34 leads by 2 months (Fig. 5a,e). Interestingly, despite stronger interannual ENSO in the MIS31 climate the correlation between the SST and precipitation indices is extremely reduced for AM, AUS and WNP monsoons during the interglacial period (Fig. 5e).

In order to verify the impact of decadal variability on the link between the NINO34 and the monsoon, Figure 5 also show the correlation between the indices but filtering out the interannual periodicity. This reveals that the decadal variability is responsable for about 40% of the correlation in the CTR climate. As should be expected, by removing the interannual frequency in the MIS31 climate, correlations values turn very small which indicate that changes in AUS, WNP and AM precipitation for this interglacial are more closely connected to hemispherical features than to the tropical-extratropical climate interaction.

## 5 Concluding Remarks

This investigation centered on a comparison between present-day conditions (CTR) and those characteristic of a super-interglacial epoch, the Marine Isotope Stage 31 (MIS31). Using coupled global climate model simulations (ICTP-CGCM), we have first demonstrated significant changes in the spatial patterns and seasonality of sea-level pressure, sea-surface tem-



peratures, and heat fluxes during the MIS31 climate compared to present-day conditions, and these changes have a significant impact on the main modes of variability. Anomalous equatorial windstress associated with a modified seasonal cycle in the MIS31 simulation leads to stronger ENSO variability compared to the present-day climate. Moreover, the decadal variability differs dramatically in the MIS31 simulation from that characteristic of present-day conditions. This decadal variability also differs greatly across the ENSO diversity spectrum, with off-equatorial atmospheric circulation playing a significant role in

inducing decadal variability.

Evaluation between paleoreconstruction and modeling results is a complex task, because reconstructions depict dominant signals in a particular time interval and locale. Thus, they cannot be assumed to geographically represent large-scale domains, and their ability to reproduce long-term environmental conditions should be considered with care.

Discrepancies between modeling results and paleoreconstructions for the MIS31 climate, which occurred under very par-

ticular conditions and high seasonality, may unfortunately be expected. The MIS31 may have been dominated for instance by vegetation patterns drastically different than today. This modifies the global evapotranspiration rates and the hydrological cycle, producing precipitation that can differ greatly from model results. This suggests that uncertainties in the model may be reduced when including more realistic boundary conditions that currently are not available.



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



**Table 1.** Averaged surface tempertures ($^{\circ}$C) for the CTR, HadCRUT4 dataset (1961-90) and ERA-I (1980-2010). Differences between MIS31 and the CTR runs are also shown. Values in brackets are for JJA (June,July and August). Otherwise values are computed for DJF (December, January and February).

| Dataset | Global | NH | SH |
|---|---|---|---|
| HadCRUT4 | 12.2 (15.7) | 8.50 (20.4) | 16.0 (11.0) |
| ERA-I | 12.6 (16.0) | 9.4 (21.0) | 16.2 (11.3) |
| CTR | 14.0 (17.3) | 10.6 (22.4) | 17.4 (12.2) |
| MIS31-CTR | -0.7 (+1.2) | -0.4 (+2.2) | -1.0 (+0.4) |

**Table 2.** Differences in reconstructed SST/LAKE E temperatures ($\Delta T_{re}$, Wet etal. (2016)) and NOAA Extended Reconstructed SST V3b/ERA-I. Differences between MIS31 - CTR SST and LAKE E temperatures in JJA ($\Delta$T). $\sqrt{}$ ($X$) stands for agreement (disagreement) between the $\Delta$T and induced SST anomalies (MIS31-CTR) induced by regressing the ENSO index. NE indicates that the index was not evaluated at the grid point or anomalies are too close to zero. Based on Wet etal. (2016) and Justino etal. (2017).

| Site (coordinates) | $\Delta T_{re}$ ($^{o}$C) Reconstruction | $\Delta T$ ($^{o}$C) Speedy-NEMO | ENSO | Reference |
|---|---|---|---|---|
| Lake E (67N 172E) | 2.5 | 1.0 | $\sqrt{}$ | (Melles etal., 2012) |
| ODP 982 (57N 15W) | 1.2 | 2.1 | $\sqrt{}$ | (Lawrence etal., 2009) |
| DSDP607 (41N 33W) | 1.7 | 2.1 | $\sqrt{}$ | (Raymo etal., 1996) |
| 306-U1313 (41N 32W ) | 2.4 | 1.9 | $\sqrt{}$ | (Naafs etal., 2013) |
| 1146 (19N 116E) | -2.6 | -1.0 | $X$ | (Herbert etal., 2010a) |
| 722 (16N 59W) | -0.9 | 1.6 | $\sqrt{}$ | (Herbert etal., 2010b) |
| 1143 (9N 113E) | -0.4 | -0.4 | $X$ | (Li etal., 2011) |
| 871 (5N 172E) | -0.4 | 1.1 | $X$ | (Dyez and Ravelo, 2014) |
| 847 (0 95W) | 2.3 | 3.0 | NE | (Medina-Elizalde etal., 2008) |
| 849 (0 110W) | 1.4 | 1.1 | NE | (McClymont and Rosell-Melé, 2005) |





**Figure 1.** a) Annual differences between MIS31 and CTR runs for SLP$_e$ and b) for Z200 (mb). c) shows differences of monthly SST ($^{\circ}$C) and windstress (vector, Nm$^{-2}$) between MIS31 and CTR runs. d) differences of thermocline depth between MIS31 and CTR simulation (meter). e) and f) show differences between MIS31 and CTR zonal windstress (Nm$^{-2}$), and precipitation differences between MIS31 and CTR runs (mm/day). g) is the Sverdrup transport in the CTR and h) the difference between the Sverdrup transport in the MIS31 and CTR runs.

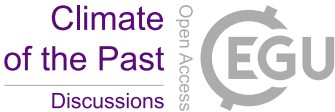



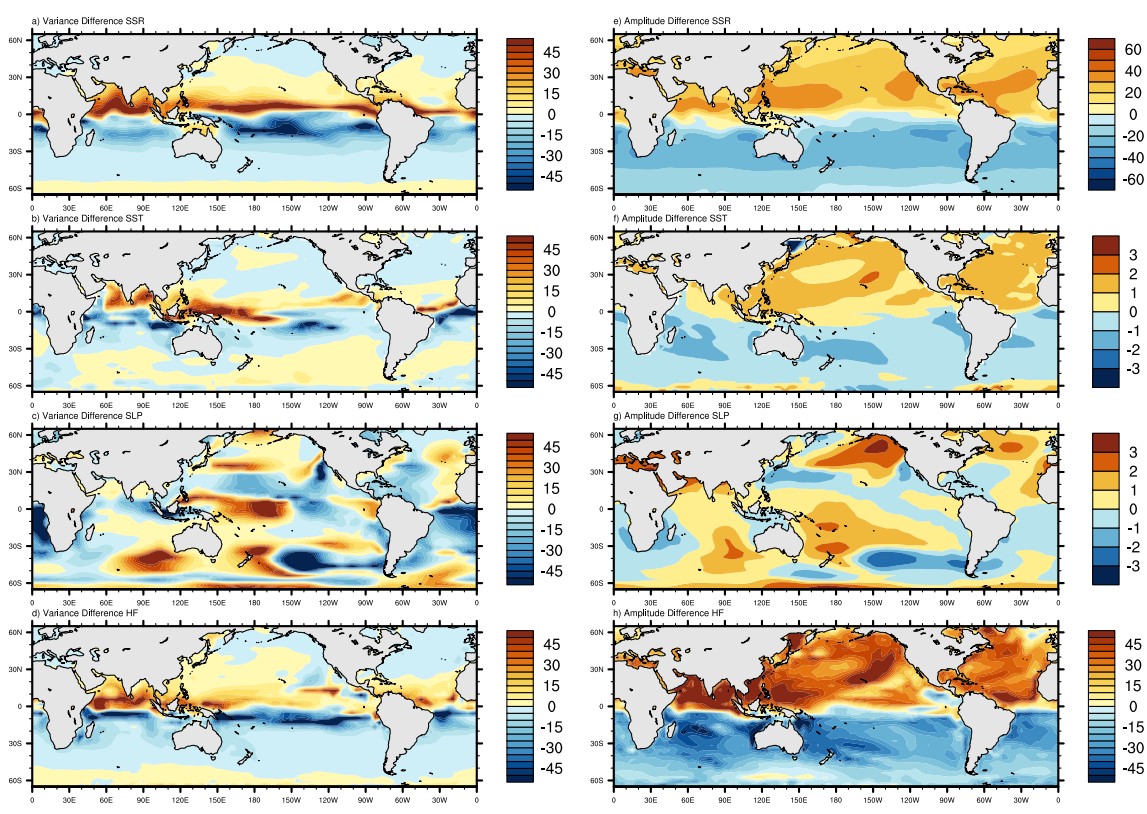

**Figure 2.** Differences of the first harmonic variance between the MIS31 and CTR run for a) surface solar radiation (W/m$^2$), b) SST ($^\circ$C), c) SLP (mb) and d) surface net heat flux (W/m$^2$). e), f), g) and h) are the same as a), b), c) and d) but for the first hamonic amplitude.





**Figure 3.** (a) MTM power spectrum of NIÑO3, NIÑO4 and NIÑO34 for the CTR run. b) is the power spectrum for the HadSST dataset. c) is in a) but for the MIS31 run. d) is the same as a) but for the windstress between 0-20°N. d)



**Figure 4.** (a) is the leading EOF of SST anomalies for the CTR simulation displayed as amplitudes (°C) by regressing hemispheric SST anomalies upon the NIÑO34 timeseries. (b) shows SST differences between the MIS31 and CTR regressed patterns. (c) and (d) are as in (a) and (b) but for SLP (mb). (e) and (f) as in (a) and (b) but for zonal wind stress ($Nm^{-2}$). Please note that Figures are shown with distinct labels





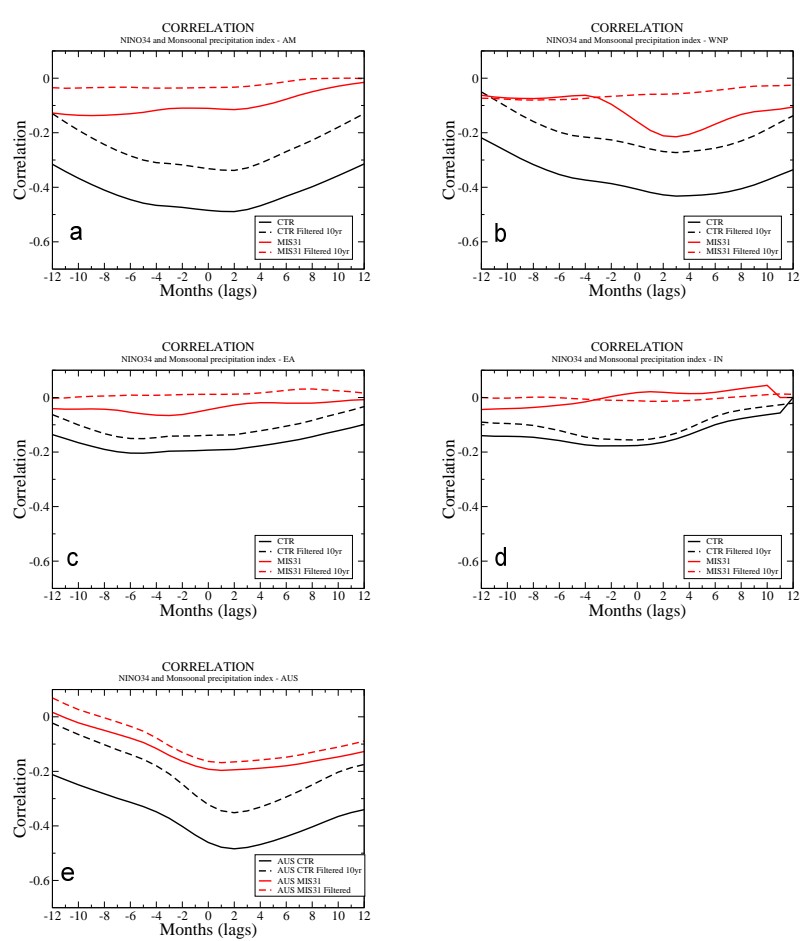

**Figure 5.** Lag correlations between the Asian and Australia monsoon domains and the NINO34 index (black for CTR and red for MIS31 simulations). (a) Asia monsoon, (b) Western North Pacific monsoon (c) East Asia monsoon (d) Indian monsoon (e) Australia monsoon