# Peer review of "A modified seasonal cycle during MIS31 superinterglacial favors stronger interannual ENSO variability"

_Climate of the Past, 2018_

## Referee Comment (RC1) · Anonymous Referee #1 · 7 Jan 2019

This study is an interesting work to be published in Climate of the Past after a moderate revision. First, I think it will be beneficial for the authors if they include "modelling study" in their title. The introduction is missing some introductory texts regarding the main message given in the title and abstract. Although I support the insightful analysis done in the manuscript, I think the authors should add few lines that why they use a certain method for their analysis and how to interpret those in terms of climate dynamics. Results in the manuscript and their implications are interesting but the main story is sometimes hidden behind. My major and minor comments regarding the text and analysis are following:

Introduction: Add few lines about the main goal of the paper and relevant previous studies

Section 2:

Line 122: which year did you use for the present-day run?

Line 126: When you talk about the difference between MIS31 and CTR, so you mean the difference between their mean over 500 years?

Line 175: eddy SLP is confusing here. I would show SLP itself as it is easier to compare it to SST and wind field. Instead, you might show eddy Z200.

Line 201: Add the latitude to the thermocline figure. You need to highlight this paragraph better as it is part of your main story.

Line 224: cold SST anomalies could be because of the displacement of Kuroshio current. Try to make this paragraph more related to the main story.

Section 3: suggested headline: Enhance seasonality in MIS31

-You need to help the reader to understand why you did harmonic analysis. Some readers may not know how to understand and interpret the analysis in Figure 2. Please explain in two/few lines. For example the 1st harmonic represents . . . and the second one represents . . ... Why not simply plotting the annual cycle to understand the change in seasonality and what is the benefit from harmonic analysis? Also instead of showing the difference between MIS31 and CTR, show the harmonic analysis for each. To avoid showing many figures, just show the analysis for the most relevant variables. Also for the variance, show it for the seasonal case. In general stay to the point.

Section 4: Restructure part of the text. First explain the stronger ENSO at interannual time scale and possible causes behind it. Recall the results from previous sections (enhanced seasonality and shallower thermocline) to link your results to each other. Then talk about the decadal variability and why it is weaker in MIS31. Again, stay to the point.

---

## Referee Comment (RC2) · Anonymous Referee #2 · 4 Feb 2019

The manuscript provides a broad overview on how modified seasonal cycle, as such from superinterglacial MIS31, could potentially impacts the variability of major climate players, as for instance the ENSO and the Monsoonal systems. The manuscript is interesting and address a timely topic, perfectly suitable to *Climate of the Past,* and it sheds some light on our understanding of future climate changes. Nevertheless, I miss some robustness in the analyses what I tried to highlight in the comments below. I believe that a major revision of the manuscript is needed, but it should be feasible taking into account what the authors have presented so far.

**Title:**
"*A modified seasonal cycle during MIS31 superinterglacial favors **stronger** ENSO variability*"
Isn't your title in conflict with your results? The authors argue that the MIS31 conditions intensify the 3-7 year (interannual) variability, while the 15-30 year (multidecadal) variability vanishes. Since the authors are addressing both bands of variability in the manuscript, I suggest they make it clear in the title what band of variability is intensified by MIS31 conditions.

**Abstract:**
The authors clearly specify that the MIS31 is characterized by "*enhanced seasonality… *" (line 2). Afterwards, it is mentioned that the MIS31 is marked by a "*weaker seasonal cycle of the wind stress*". So, not all climate players have an "enhanced seasonality"(?). Maybe it is worth to add few words in order to explain what parameters (climate features) have the seasonal cycle intensified. At this stage, things are not clear in the abstract

**1. Introduction**:

**General comment:** The authors presented a convincing story to explain how the enhanced seasonal cycle from the interglacial MIS31 can be used as a proxy for understanding the potential impact of increased atmospheric CO2 in the climate players, as for instance the ENSO. Nevertheless, in my opinion, the introduction could also make it clear the main and specific scientific questions that the authors are going to address. As it is, things are a bit vague.

**line 4, pg. 2:** "*… temperatures that were **several** degrees...*": a number would be helpful.

**line 11, pg. 2:** "*distubances*" → disturbances

**line 21, pg. 2:** "*Yin etal. (2014) indicates*" → indicate

**lines 23-24, pg. 2:** "*… warmer **conditions** during the MIS13, …, **amplifies** … and **contributes***" → amplify, contribute

**line 26, pg. 2:** "*… sea surface **temperatures** (SSTs)… **contributes***" → contribute

**line 30, pg. 2:** "**(Sun etal., 2010b) based** on …**,** demonstrated...*" → Sun etal. (2010b), …, demonstrated… Also, this sentence sounds a bit confusing. For instance, "*… based on seven million years of wind and precipitation variability*" sounds like a variability band of 7 million years. Also, "*monsoonal **fluctuations**… is*" → are. Please, reconsider to rephrase it.

**line 7, pg. 2: "*The effect** of ocean dynamics also modify...*"** → modifies

**line 13, pg. 3:** "distict" → distinct

**2. Coupled Climate Simulations**

**line 28, pg. 3:** It would be nice if the authors mention here what are the source of the "*present-day boundary conditions*".

**line 29, pg. 3:** Missing brackets ')'. Also, it isn't clear to me the link with "*Fig. 1 of supplementary material by Justino etal. (2017)*". This figure shows the MIS31 WAIS topography and the differences of incoming solar radiation between CRT and MIS31 simulations. Are these the only two differences between the CRT and MIS31 runs? I recommend  the authors make it clearer all differences between both experiments. I think it is a bit boring to the reader search for a key information in another manuscript, but this is only my personal opinion and I leave to the authors to decide whether to incorporate a relevant figure to this manuscript as well.

**lines 29-31, pg. 3:** The experiments were run to 2000 (1000) years to equilibrium and the analyses were based on the last 500 years. What are the total time spans for each run: 2500 and 1500 years?

**lines 5-6, pg. 4:** "*… but a brief discussion of the … are provided below*" → is provided

**line 7, pg. 4:** Define HadCRUT4

**lines 7-15, pg. 4; Table 1:** The comparison among averages is much more meaningful if followed by the respective standard deviations. The values can be similar (as the authors argue for CRT and ERAI), but global and hemispheric averages can hide important regional differences. I think global maps of mean and std temperature for each of the products (HadCRUT4, ERA-I, CRT, and MIS31) would provide a much more complete assessment on the differences/similarities among them. The authors could add such a figure to the supplementary material.

Also, I was puzzled by the amplitude of the seasonal cycle in the Southern Hemisphere. See the table below, which is based on the values from manuscript's Table 1.

|                   | HN, CTR | HN, MIS31 | HS, CTR | HS, MIS31 |
|-------------------|---------|-----------|---------|-----------|
| **Summer**        | 22.4    | 24.6      | 17.4    | 16.4      |
| **Winter**        | 10.6    | 10.2      | 12.2    | 12.6      |
| **Summer - Winter** | 11.8  | 14.4      | 5.2     | 3.8       |

If we (simply) assume that the amplitude of the seasonal cycle is the difference between summer and winter averages, the Southern Hemisphere shows larger amplitude during CRT conditions (5.2 C) compared to MIS31 conditions (3.8 C). Could the authors comment on that and explain why such a difference happens? Is the enhanced amplitude of the MIS31 seasonal cycle expected only in the Northern Hemisphere? Also, being the MIS31 a super-interglacial, is there a reason to explain why the summer temperatures are higher during CRT conditions (17.4 C) compared to the MIS31 conditions (16.4 C) in the Southern Hemisphere?

**lines 16-23, pg. 4:** Still related to the comment above, this paragraph would be more complete with the suggested global maps.

**line 17, pg. 4:** "*… differences between the MIS31 and CTR simulation… *" → simulations

**lines 25 and 28, pg. 4; also in other parts of the manuscript:** "(not shown)". For 5 times in the manuscript the authors use "not shown". Maybe the authors should consider to show some of the "not shown" results.

**line 29, pg. 4:** Consider to define SLP. It may help a non-specialized reader.

**lines 9-10, pg. 5:** I like the analysis regarding the changes in the wind and the resulting equatorial upwelling. Maybe an improved analysis in terms of Ekman Transport and Ekman Pumping would improve further the manuscript. Also, as Fig. 1c is plotted (scales and spacing between wind vectors), sometimes is hard to compare the text against the results. The authors could consider to plot also the differences in the wind stress curl or alternatively the differences of vertical Ekman Pumping velocities – $W_E$. In my view this is an important find of the manuscript and deserves more attention.

**line 13, pg. 5:** Levitus etal. (2010) is fine but I would suggest a more up to date product, as for instance the World Ocean Database 2018 (WOD18).

**lines 9-10, pg. 5:** I also like the approach the authors used by applying the Sverdrup conceptual model in order to inspect the changes in the subtropical gyre. This is a straightforward and elegant way to look at the changes in the poleward transport at the continent's western boundaries. I would just add a line to explain that even though the wind grid-resolution is coarser than the horizontal scale of the western boundary current (ie, the Kuroshio Current), the Tx used in the calculation is a representation of the zonal-averaged wind stress so that it is still fine for this analysis. But, I leave it to the authors.

**3. Harmonic analysis of MIS31 and CTR climates**

General comment: As already mentioned by the other reviewer, I also missed a better explanation on why the authors are using the proposed methodology. I reinforce that this should be a major point to be addressed.

**line 2, pg. 6:** "The first order harmonics of meteorological parameters show long-term effects...". Only meteorological? You are also applying this analysis to SST (Fig. 2c-d).

**line 12, pg. 6:** Define HF.

**line 20, pg. 6:** *"(2b-d)"*. 2b-c?

**line 20, pg. 6:** *"displyed" → displayed*

I miss a discussion regarding Fig. 2f.

**4. MIS31 – Temporal and spatial characteristics of ENSO**

**lines 12-13, pg. 7:** Isn't clear why the authors are using the HadISST data (I guess to have an observational reference). If so, that is fair and appreciate. Please, clarify.

**lines 13-15, pg. 7:** *"This is achieved by applying the MTM… fill limitations of conventional Fourier analyses"* Since the authors mentioned it, I think it is worth to specify what are these limitations.

**from line 16 pg. 7, also pgs. 8 and 9:** Fig. 3 should display the significance levels (95%, for instance). It is hard to evaluate the authors analysis without these information. For instance, I can't exactly spot the significant bands of variabilities in Fig. 3. We do see the peaks, but Fig. 3c is marked by broad band and not necessarily the entire band is over the significance level. Also,

further information is missed on the preparation of the time series before applying the MTM. Are the time series detrended and/or normalized?

**line 20, pg. 7:** *"… attributes"* → attributes

**lines 21-22, pg. 7:** *"It is interesting to note that… weakest in NINO4"*. Do the authors have an explanation for that? Also, as suggested by Fig. 3a, the spectrum of NINO4 is shifted to higher frequencies compared to the other two indeces. Please, clarify.

**lines 22-23, pg. 7:** *"The HadISST does not show any periodicity on decadal time scales… **the length of the timeseries does not seem to capture this lower frequency**"*. This explanation does not sound convincing. A time series with 147 years (1870 to 2016) should be long enough to capture several cycles (147/30≈5; 147/15≈10). It seems that the multidecadal variability is not present in HadISST. Could the authors explain the potential reason for that? It may be due to the coarse data distribution (*in situ* observations) until the incorporation of satellite data (1982) to this product?

**line 24, pg. 7:** *"claims"* → claim

**lines 24-25, pg. 7:** *"… zonal asymmetry related to the decadal variability in the HadISST observations is weaker and not as regular as for instance in the ECHO-G model"*. Again, maybe this absence could be justified by the few data available (in spatial and temporal terms) used in the optimal interpolations for the pre-satellite period. This is just a speculation that the authors could confirm (or not ?) by searching in the literature. It is a bit confusing to mention ECHO-G, since this model wasn't referred before. If this info is really important, please provide further information.

**line 32, pg. 7:** *"This simulation shows stronger power spectrum at interannual time scales 3-7"*. As mentioned above, this statement needs to be corroborated by the confidence levels in Fig. 3.

**line 37 pg. 7:** Define SOI.

**lines 4-5 pg. 8:** *"This is in line… enhanced power also at interdecadal time scales (Fig. 3d)"*. The control run shows spectral peak both at the interannual and multidecadal timescales (Fig. 3d). In the text the authors have discussed a potential reason of why the multidecadal peak isn't observed in the MIS31 run. Nevertheless, the spectrum also doesn't show a peak for the interannual variability. Do the authors have an answer for that? This is an important point that should be addressed.

**Caption of Table 1:** *"1961-90"* → 1961-1990; Also, "June,July" → June, July.

---

## Author Comment (AC2) · 1 Mar 2019

**Dear Referee #2**

We would like to thank the reviewer for the valuable comments to improve the quality of our manuscript. These comments effectively clarified the analyses and embedded the results in a small window for misinterpretation. Please find enclosed a point-by-point reply to the reviewer' comments and suggestions.

**Answer to reviewer comments in BLUE**

*Title*

The title has been modified:

A modified seasonal cycle during MIS31 superinterglacial favors stronger **interannual** ENSO variability

*Abstract*

"*weaker seasonal cycle of the wind stress*"

We have removed the sentence above from the Abstract to avoid misunderstanding.

*Introduction*

*General comment*

Similar comment has been raised by the reviewer#1. We believe that the new Introduction covers and explores more properly the issues investigated in the Manuscript. It also points out the need for understanding in more details relevant climate mechanisms that have been overlooked in previous publications.

**line 4, pg. 2:** "*... temperatures that were **several** degrees...*": a number would be helpful.

This has been modified:

This interval was characterized by boreal summer temperatures that were several degrees greater than modern climate (up to 6C), with a substantial recession of the Northern Hemisphere (NH) sea ice \citep{melles,justino2017}.

**line 11, pg. 2:** "*distubances*" → disturbances

Modified

**line 21, pg. 2:** "*Yin etal. (2014) indicates*" → indicate

Modified

**lines 23-24, pg. 2:** "*... warmer **conditions** during the MIS13, ..., **amplifies** ... and **contributes***" → amplify, contribute

Modified

**line 26, pg. 2:** "*... sea surface **temperatures (SSTs)... contributes***" → contribute

Modified

**line 30, pg. 2:** "**(Sun etal., 2010b) based** on ...**, demonstrated...**" → Sun etal. (2010b)

This paragraph has been modified as shown below:

Based on paleo-reconstruction of wind and precipitation on the Chinese Loess Plateau, \citep{sun2010seven} demonstrated that monsoonal fluctuations at orbital-to-millennial scales is dynamically linked to changes in solar insolation, and internal boundary conditions. Therefore, it can be assumed that changes in insolation or increased temperatures as occurred during interglacial stages may trigger a distinct pattern of global monsoon, likewise can be expected in the future \citep{hsu2102}.

**line 7, pg. 2: "*The effect* of ocean dynamics also modify...**"** → modifies **line 13, pg. 3:** "distict" → distinct

This paragraph has been removed to avoid misunderstanding.

**2. Coupled Climate Simulations**
**line 28, pg. 3:** It would be nice if the authors mention here what are the source of the "*present-day boundary conditions*".

**line 29, pg. 3:** Missing brackets ')'. Also, it isn't clear to me the link with "*Fig. 1 of supplementary material by Justino etal. (2017)*". This figure shows the MIS31 WAIS topography and the differences of incoming solar radiation between CRT and MIS31 simulations. Are these the only two differences between the CRT and MIS31 runs? I recommend the authors make it clearer all differences between both experiments. I think it is a bit boring to the reader search for a key information in another manuscript, but this is only my personal opinion and I leave to the authors to decide whether to incorporate a relevant figure to this manuscript as well.

**lines 29-31, pg. 3:** The experiments were run to 2000 (1000) years to equilibrium and the analyses were based on the last 500 years. What are the total time spans for each run: 2500 and 1500 years?

We have included in the revised MS the paragraph below:

Two simulations are evaluated: a modern climate driven by present-day boundary conditions (CTR) and a second experiment for the MIS31 forcing. The CTR simulation was run to equilibrium for 2000 years, and our modern climate is the time average of the last 500 years of the CTR simulation. The CTR is run under present day orbital forcing and $CO_2$ concentration of 325 ppm as it characterizes emission by the year 1950. The

MIS31 run starts from equilibrated CTR conditions, including modifications of the WAIS topography based on \citet{pollardnature}, and the planetary astronomical configuration of 1.072 Ma according to \citet{coletti}. It has been carried out for 1000 years and the analyses take into account the last 500 years of the simulation.

The implementation of MIS31 Antarctic topography differs from the CTR counterpart primary by the absence of the WAIS, which according to \citet{pollardnature}, was induced by changes in ocean melt via the effect on ice-shelf buttressing that coincides with strong boreal summer insolation anomalies. In all experiments, the $CO_2$ concentration was set to 325 ppm which is based on boron isotopes in planktonic foraminifera shells for the MIS31 interval \citep{Honisch}.

**lines 5-6, pg. 4:** "*... but a brief discussion of the ... **are** provided below*" $\rightarrow$ is provided

**Modified**

**line 7, pg. 4:** Define HadCRUT4

We have noted that our discussion in the previous version about observed SST was based on NOAA Extended Reconstructed SST V3b instead of HadCRUT4. In the current revised version, however, we have removed all discussion involving the NOAA SST data, but kept the comparison of the CTR run with ERA-I.

**lines 7-15, pg. 4; Table 1:** The comparison among averages is much more meaningful if followed by the respective standard deviations. The values can be similar (as the authors argue for CRT and ERAI), but global and hemispheric averages can hide important regional differences. I think global maps of mean and std temperature for each of the products (HadCRUT4, ERA-I, CRT, and MIS31) would provide a much more complete assessment on the differences/similarities among them. The authors could add such a figure to the supplementary material.

The suggested figure is shown at the supplementary material and shown below.

[Figure]

Fig. 1 Supp. Material. Time averaged surface temperature for ERAI (top right), the CTR (middle right) and the MIS31 simulation (bottom right). Top, middle and bottom left are the standard deviation delivered by the datasets.

As discussed in the MS, the ICTP-CGCM is able to reproduce the main features of global temperatures insofar as time averaged is concerned. The ICTP-CGCM performs fairly in reproducing the monthly variability of temperatures as shown by the standard deviation (STD). It is demonstrated that higher values are over Asia and North America primary related to the high seasonality associated with the landmass. Larger values are also observed over oceanic regions along storms preferential track. However, due to the model resolution, limitation is noted over steep topographies such as Tibet plateau, Andes and Rocky mountain.

These considerations have been included in the revised MS.

Also, I was puzzled by the amplitude of the seasonal cycle in the Southern Hemisphere. See the table below, which is based on the values from manuscript's Table 1.

If we (simply) assume that the amplitude of the seasonal cycle is the difference between summer and winter averages, the Southern Hemisphere shows larger amplitude during CRT conditions (5.2 C) compared to MIS31 conditions (3.8 C). Could the authors comment on that and explain why such a difference happens? Is the enhanced amplitude of the MIS31 seasonal cycle expected only in the Northern Hemisphere? Also, being the MIS31 a super-interglacial, is there a reason to explain why the summer temperatures are higher during CRT conditions (17.4 C) compared to the MIS31 conditions (16.4 C)
in the Southern Hemisphere?

This is a very interesting point raised by the reviewer and certainly needs clarification.
Figure 2 (Supp. Material) shows the monthly averaged hemispheric pattern for surface
solar radiation (SSR) and surface temperatures delivered by the MIS31 and CTR
simulations. This figure demonstrates an inter-hemispheric seesaw emphasizing the
substantial increase in the boreal SSR during the summer season in the MIS31
experiment, and similar situation occurs in the Southern Hemisphere during DJF in the
extra-tropics. It has to be argue that the reason for larger seasonality in the SH is related
to the excess of SSR in DJF but deficit in JJA as compared to the NH (Fig 2a,b Supp.
Material). Thus, much warmer summer conditions and colder winter/spring in the SH
increase the annual amplitude.

We have to make clear that according to Table 1 *"summer temperatures are NOT higher*
*during CTR conditions (17.4 C) compared to the MIS31 conditions (16.4 C) in the*
*Southern Hemisphere"*. In fact, the summer values in Table 1 are in brackets: in the NH
(SH) the MIS31 is 2.2C (0.4C) warmer than CTR simulation. In the SH MIS31 is 0.4C
warmer.

[Figure]

[Figure]

Figure 2 Supp. Material. a) Zonally averaged surface solar radiation for the MIS31 and CTR simulations. b) The same as in a) but for surface temperatures.

**lines 16-23, pg. 4:** Still related to the comment above, this paragraph would be more complete with the suggested global maps.

Modified

**line 17, pg. 4:** *"... differences between the MIS31 and CTR simulation... "* →
simulations

Modified

**lines 25 and 28, pg. 4; also in other parts of the manuscript:** "(not shown)". For 5 times in the manuscript the authors use "not shown". Maybe the authors should consider to show some of the "not shown" results.

We have included in the Supp. Material additional figures useful to clarify the MS results.

**line 29, pg. 4:** Consider to define SLP. It may help a non-specialized reader.

Included

**lines 9-10, pg. 5:** I like the analysis regarding the changes in the wind and the resulting equatorial upwelling. Maybe an improved analysis in terms of Ekman Transport and Ekman Pumping would improve further the manuscript. Also, as Fig. 1c is plotted (scales and spacing between wind vectors), sometimes is hard to compare the text against the results. The authors could consider to plot also the differences in the wind stress curl or alternatively the differences of vertical Ekman Pumping velocities – W. In my view this is an important find of the manuscript and deserves more attention.

We agree with the reviewer that plotting the vertical velocities would bring benefits to the article. However, changes in the thermocline depth (Fig. 1d) is very much related to upwelling, vertical velocity and modifications in the sub-tropical cell, therefore similar results may arise from the calculation of Ekman dynamics. Figure 1c shows the wind anomalies between MIS31 and CTR simulations.

**line 13, pg. 5:** Levitus etal. (2010) is fine but I would suggest a more up to date product, as for instance the World Ocean Database 2018 (WOD18).

It is shown below the thermocline depth for Levitus (top left), GLORYS reanalysis from 1993-2015 (top right) and ICTP-CGCM (bottom). Based on these plots we note that no large differences appear between the reanalyses (Levitus and GLORYS) and the ICTP-CGCM. Our CTR climate, however, shows a much shallow thermocline off the equatorial region in the SH.

[Figure]

**lines 9-10, pg. 5:** I also like the approach the authors used by applying the Sverdrup
conceptual model in order to inspect the changes in the subtropical gyre. This is a
straightforward and elegant way to look at the changes in the poleward transport at the
continent's western boundaries. *I would just add a line to explain that even though the*
*wind grid-resolution is coarser than the horizontal scale of the western boundary*
*current (ie, the Kuroshio Current), the Tx used in the calculation is a representation of*
*the zonal-averaged wind stress so that it is still fine for this analysis.* But, I leave it to
the authors.

We have included the suggested statement.

**3. Harmonic analysis of MIS31 and CTR climates**

General comment: As already mentioned by the other reviewer, I also missed a better
explanation on why the authors are using the proposed methodology. I reinforce that
this should be a major point to be addressed.

We have included the following paragraph to describe in more details the choice for using
harmonic analyses.

The use of harmonic analysis allows the identification of dominant climate signals in the
space–time domain, separating small and high frequency processes (e.g diurnal cycle)
from large-scale features (e.g. seasonal). Analyses conducted on the frequency domain
can capture and differentiate the contribution of all time-scales. Thus, different climate
regimes and transition regions can be characterized. The 1st harmonic shows the
dominance of the annual cycle when most of the variance is represented by this harmonic.
It has to be stressed that investigations based upon area averaged time series are
embedded with small and large-scale processes dictated by distinct periodicity, this in
turn hampers the identification of periodic climatic signals in the space–time domain
\citep{justino-ijoc,cli4010003}.

**line 2, pg. 6:** "The first order harmonics of meteorological parameters show long-term
effects...". Only meteorological? You are also applying this analysis to SST (Fig. 2c-d).

Modified

**line 12, pg. 6:** Define HF.

Defined

**line 20, pg. 6:** *"(2b-d)"*. 2b-c?

2b-c is correct. It has been modified in the MS.

**line 20, pg. 6:** *"displyed"* → *displayed*

I miss a discussion regarding Fig. 2f.

It was discussed but the Figure was not cited.

**4. MIS31 – Temporal and spatial characteristics of ENSO**

**lines 12-13, pg. 7:** Isn't clear why the authors are using the HadISST data (I guess to have an observational reference). If so, that is fair and appreciate. Please, clarify.

As mentioned previously we have removed all comments and discussion on the HadISST

**lines 13-15, pg. 7:** *"This is achieved by applying the MTM... fill limitations of conventional Fourier analyses"* Since the authors mentioned it, I think it is worth to specify what are these limitations.

The limitation we were referring to is because the Fourier formulation assumes that the individual coefficient represents the amplitude and phase of the corresponding frequency. We have inserted additional information of the MTM approach.

Since the statement below does not contribute to the paper results it is not included in the revised version (…*fill limitations of conventional Fourier analyses···*).

**from line 16 pg. 7, also pgs. 8 and 9:** Fig. 3 should display the significance levels (95%, for instance). It is hard to evaluate the authors analysis without these information. For instance, I can't exactly spot the significant bands of variabilities in Fig. 3. We do see the peaks, but Fig. 3c is marked by broad band and not necessarily the entire band is over the significance level.

The new Figure 3 provides the significance levels 99, 95 and 90%.

Also, further information is missed on the preparation of the time series before applying the MTM. Are the time series detrended and/or normalized?

It has been included in the revised MS.
This is achieved by applying the Multi-Taper method to detrended timeseries, 3 tapers have been used to resolve spectral fluctuations at frequencies greater than the Rayleigh frequency \citep[MTM; ][]{thomson}.

**line 20, pg. 7:** *"... attributes"* →attributes

**lines 21-22, pg. 7:** *"It is interesting to note that... weakest in NINO4"*. Do the authors
have an explanation for that?

The weakening of decadal variability in the NI\~NO4 region may be related to wind
variability in the off-equatorial tropics as proposed by \citet{nonaka}. This assumption
has been verified by computing the correlation pattern associated with the NI\~NO
indices. It turns out that the NI\~NO4 relationship with the zonal windstress within 10-
30$^\circ$N is considerably weaker than that of NI\~NO34 or NI\~NO3. Moreover, this
weaker correlation between the NI\~NO4 and windstress is not confined to the equatorial
region but extends to the extratropics.

Also, as suggested by Fig. 3a, the spectrum of NINO4 is shifted to higher frequencies
compared to the other two indeces. Please, clarify.

The reason to this slightly shift to higher frequency by the NINO4 is not clear, however,
because the NINO4 is located much closer to the warming pool region, which is
dominated by weak seasonal cycle with the 1st harmonic explaining by about 30% of the
total variance, may indicate that higher order harmonics play a role to induce some power
at higher frequency. The NINO4 power spectrum in the MIS31 run does not show
dominant periodicity at interannual and interdecadal timescales.

This is included in the MS.

**lines 22-23, pg. 7:** *"The HadISST does not show any periodicity on decadal time*
*scales... the length of the timeseries does not seem to capture this lower frequency"*.
This explanation does not sound convincing. A time series with 147 years (1870 to
2016) should be long enough to capture several cycles (147/30≈5; 147/15≈10). It seems
that the multidecadal variability is not present in HadISST. Could the authors explain
the potential reason for that? It may be due to the coarse data distribution (*in situ*
observations) until the incorporation of satellite data (1982) to this product?

As mentioned previously we have removed discussion on HadISST.

**line 24, pg. 7:** *"claims"* → claim

Modified

**lines 24-25, pg. 7:** "*... zonal asymmetry related to the decadal variability in the*
*HadISST observations is weaker and not as regular as for instance in the ECHO-G*
*model*". Again, maybe this absence could be justified by the few data available (in
spatial and temporal terms) used in the optimal interpolations for the pre-satellite
period. This is just a speculation that the authors could confirm (or not ?) by searching
in the literature. It is a bit confusing to mention ECHO-G, since this model wasn't
referred before. If this info is really important, please provide further information.

Removed

**line 32, pg. 7:** *"This simulation shows stronger power spectrum at interannual time scales 3-7".* As mentioned above, this statement needs to be corroborated by the confidence levels in Fig. 3.

Confidence levels have been show.

**line 37 pg. 7:** Define SOI.

It has been defined.

**lines 4-5 pg. 8:** *"This is in line... enhanced power also at interdecadal time scales (Fig. 3d)".* The control run shows spectral peak both at the interannual and multidecadal timescales (Fig. 3d). In the text the authors have discussed a potential reason of why the multidecadal peak isn't observed in the MIS31 run. Nevertheless, the spectrum also doesn't show a peak for the interannual variability. Do the authors have an answer for that? This is an important point that should be addressed.

The MIS31 climate shows dominant power spectrum for the NINO3 and NINO34 at inter-annual timescales distributed at a broader 3-12 year band, differing from the CTR that exhibits a shorter band, 6-8 years. According to Feldstein (2000) the power spectrum is defined by the interannual variance due to external forcing and the interannual variance from stochastic processes. The power spectrum which is dominated by the external forcing exhibits a sharper peak as compared to that driven by stochastic processes. It may be argued that despite the dominance of external forcing in the MIS31 climate random processes also play a significant role to define the temporal variability inducing the broader frequency band as compared to the CTR climate.

**Caption of Table 1:** *"1961-90"* → 1961-1990; Also, "June,July" → June, July.

Modified as suggested.

---

## Author Comment (AC3) · 4 Mar 2019

**A modified seasonal cycle during MIS31 superinterglacial favors stronger interannual ENSO variability**

Flavio Justino[1], Fred Kucharski[2], Douglas Lindemann[1], Aaron Wilson[3], and Frode Stordal[4]

[1] Department of Agricultural Engineering, Universidade Federal de Vicosa, PH Rolfs, Vicosa, Brazil
[2] The Abdus Salam International Centre for Theoretical Physics, Trieste, Italy
[3] Polar Meteorology Group, Byrd Polar and Climate Research Center, The Ohio State University, Columbus, OH, USA
[4] University of Oslo,Department of Geosciences, Forskningsparken Gaustadalleen, Oslo, Norway

Correspondence:  Flavio Justino (fjustino@ufv.br)

[Figure]

Figure 1 Time averaged surface temperature for ERAI (top right), the CTR (middle right) and the MIS31 simulation (bottom right). Top, middle and bottom left are the standard deviation delivered by the datasets.

[Figure]

Figure 2 a) Zonally averaged surface solar radiation for the MIS31 and CTR simulations. b) The same as in a) but for surface temperatures.

5

10

[Figure]

[Figure]

Figure 3. (a) MTM power spectrum of windstress between 0-20N for the CTR simulation and b) is the same but for the MIS31 simulation. Red, green and blue lines show the 90%, 95% and 99% significance levels.

5

---

## Author Comment (AC4) · 4 Mar 2019

ENSO power spectrum showing the 90, 95 and 99% significance levels

[Figure]

**Fig. 1.**

---

## Author Comment (AC1)

**Dear Editor and Referee #1**

We would like to thank the reviewer for the valuable comments to improve the quality of our manuscript. These comments effectively clarified the analyses and embedded the results in a small window for misinterpretation. Please find enclosed a point-by-point reply to the reviewer' comments and suggestions.

**Answer to reviewer comments in BLUE**

**Reviewer primary comments –**

- *"The introduction is missing some introductory texts regarding the main message given in the title and abstract."*

The revised version of the Manuscript (MS) exhibits a modified **Introduction** in order to address in more details the issues investigated. It begins covering the importance of ENSO and equatorial Pacific for the global climate in distinct eras. The new **Introduction** also includes discussion of previous studies on the relationship between the ENSO and the monsoonal system. We finalize the **Introduction** exploring the importance of understanding the ENSO, the equatorial Pacific and the monsoon system during interglacial stages to shed light on the potential effect of future human-induced climate change.

- *"I think the authors should add few lines that why they use a certain method for their analysis, in particular harmonic analyses)"*

We have provided in the revised MS a better explanation of harmonic analyses as well as put forward advantages of using this approach to explore the magnitude of the annual cycle, as shown below:

We have included the following paragraph to describe in more details the choice for using harmonic analyses.

The use of harmonic analysis allows the identification of dominant climate signals in the space–time domain, separating small and high frequency processes (e.g diurnal cycle) from large-scale features (e.g. seasonal). Analyses conducted on the frequency domain can capture and differentiate the contribution of all time-scales. Thus, different climate regimes and transition regions can be characterized. The 1st harmonic shows the dominance of the annual cycle when most of the variance is represented by this harmonic. It has to be stressed that investigations based upon area averaged time series are embedded with small and large-scale processes dictated by distinct periodicity, this in turn hampers the identification of periodic climatic signals in the space–time domain \citep{justino-ijoc,cli4010003}.

- *"Results in the manuscript and their implications are interesting but the main story is sometimes hidden behind"*

The revised MS provides much deeper discussion on the results exploring the question and comments of both reviewers. Moreover, we have provided additional figures as Supplemetary Material. Those figures are shown in the document which includes responses to the reviewer #2.

**SECTION 2**

*Line 122: which year did you use for the present-day run?*

The paragraph below has been included in the revise MS:

Two simulations are evaluated: a modern climate driven by present-day boundary conditions (CTR) and a second experiment for the MIS31 forcing. The CTR simulation was run to equilibrium for 2000 years, and our modern climate is the time average of the last 500 years of the CTR simulation. The CTR is run under present day orbital forcing and $CO_2$ concentration of 325 ppm as it characterizes emission by the year 1950. The MIS31 run starts from equilibrated CTR conditions, including modifications of the WAIS topography based on \citet{pollardnature}, and the planetary astronomical configuration of 1.072 Ma according to \citet{coletti}. It has been carried out for 1000 years and the analyses take into account the last 500 years of the simulation.

The implementation of MIS31 Antarctic topography differs from the CTR counterpart primary by the absence of the WAIS, which according to \citet{pollardnature}, was induced by changes in ocean melt via the effect on ice-shelf buttressing that coincides with strong boreal summer insolation anomalies. In all experiments, the $CO_2$ concentration was set to 325 ppm which is based on boron isotopes in planktonic foraminifera shells for the MIS31 interval \citep{Honisch}.

*Line 126: When you talk about the difference between MIS31 and CTR, so you mean the difference between their mean over 500 years?*

Yes, it is. This has been clarified in the revised MS.

*Line 175: eddy SLP is confusing here. I would show SLP itself as it is easier to compare it to SST and wind field. Instead, you might show eddy Z200.*

We have shown the $SLP_e$ because differences between high and low pressure dominant features in the MIS31 and CTR are enhanced, such as at the subtropical N. Pacific and Azores high, the Aleutian low. This facilitates the interpretation of wind anomalies at the subtropics and equatorial region (e.g the trade wind anomalies). It is shown below, for your consideration, the SLP differences between the two runs, where is noted very similar pattern as delivered by the $SLP_e$ presented in Figure 1a. We have included the text above in the revised MS.

[Figure]

SLPMIS31-SLP

*Line 201: Add the latitude to the thermocline figure. You need to highlight this para-*
*graph better as it is part of your main story.*

The new thermocline figure includes the latitude labels.

We have added to the revised MS a discussion on the role of the thermocline to
characterize the ENSO phase and amplitude, as below:

Modification in the near surface atmospheric circulation can also modify the oceanic
vertical characteristics affecting the thermocline depth and ENSO
\citep{wen2014,bush01}. As discussed by \citet{yang2009} for the equatorial Pacific,
changes in the depth of the thermocline determines the SST magnitude and the behavior
of the air-sea interaction, influencing the phase, amplitude, and time scale of the tropical
climate.

  The ICTP-CGCM properly reproduces the equatorial thermocline depth (using the depth
of maximum vertical temperature gradient) compared to the Levitus dataset
\citep{Levitus} and to GLORYS reanalysis. The MIS31 forcing leads to a shallower
thermocline and reduction of its zonal gradient (Fig. \ref{fig1}d), which is primarily
related to the anomalous wind flow \citep[e.g., ][]{zebiak86,an1999}.

  A deeper thermocline however, is observed in part of the NI\~NO3 region (Fig.
\ref{fig1}d, contour). In the eastern Pacific, thermocline dynamics have been associated
with changes in SST, the air-sea coupling, and ENSO \citep{leduc,yang2009}. This
implies a weaker Walker circulation during the MIS31 interval that is supported by SST
reconstructions (from Ocean Drilling Program sites 849, 847, 846, and 871) in the western
and eastern equatorial Pacific \citep{clymont}.

*Line 224: cold SST anomalies could be because of the displacement of Kuroshio cur-*
*rent. Try to make this paragraph more related to the main story.*

The negative SST anomalies are primary located in the warming pool region (10S-20N)
and reach only the south-most part of the Kuroshio current (Figure below). Therefore, we

argue that the intensification of the trade winds, local upwelling and the evaporative feedback should play the main role in leading the anomalous SST pattern.

[Figure]

*Section 3: suggested headline: Enhance seasonality in MIS31*

The section title has been modified.

---

## Author Response (AR1)

**Dear Editor**

We would like to thank you for your kindness in handling the manuscript and the reviewer for the valuable comments to improve the quality of our manuscript. All suggested changes have been incorporated in the manuscript, following the sequence in our response to reviewers.

**Best Regards,**

**Flavio Justino**

---

## Author Response (AR2)

**Dear Editor**

We would like to thank you for valuable comments to improve the quality of our manuscript. Our modified paper has tracked changes in RED, according to the response to reviewers. It has to be mentioned that both response to reviewers documents and supplementary material have already been uploaded.

Best Regards

Flavio Justino

---

## Author Response (AR3)

**Dear Editor**

We would like to thank you for valuable comments to improve the quality of our manuscript. The present version of our Manuscript includes the 3 corrections requested by the Editor as well as changed the reference Yin and Berger, 2011 to 2012.

Editor remarks:

-Page 10 ,line 14: (Fig.5a) during this interglacial (An, 2000, Sun et al., 2010a)?
-Page 10, line 20: it shows clearly the weakening of ....?
-Page 10, line 26: during this interglacial period

We kindly ask to modify the paper title as suggested by the Reviewer #2 to:

**A modified seasonal cycle during MIS31 superinterglacial favors stronger interannual ENSO and monsoon variability**

Best Regards

Flavio Justino